# Design and Experimental Validation of a 3-DOF Force Feedback System Featuring Spherical Manipulator and Magnetorheological Actuators †

**Bao Tri Diep [1,2], Ngoc Diep Nguyen [2], Thanh T. Tran [3] and Quoc Hung Nguyen [2,3,*]**

1 Faculty of Civil Engineering and Applied Mechanics, HCMC University of Technology and Education, Ho Chi Minh City 71307, Vietnam; dtridb.ncs@hcmute.edu.vn
2 Faculty of Mechanical Engineering, Industrial University of Ho Chi Minh City, Ho Chi Minh City 71408, Vietnam; nguyenngocdiep@iuh.edu.vn
3 Faculty of Engineering, Vietnamese-German University, Binh Duong Province 75114, Vietnam; thanh.tt@vgu.edu.vn
* Correspondence: hung.nq@vgu.edu.vn; Tel.: +(84)938485812
† This paper is an extended version of the conference paper "Development of 3-DOF Force Feedback System Using Spherical Arm Mechanism and MR Brakes". In Proceedings of the 3rd European Conference on Materials, 16-18 February, 2019, Amsterdam, Netherlands.

**Abstract:** This research focuses on the development of a new 3-DOF (Degree of Freedom) force feedback system featuring a spherical arm mechanism and three magnetorheological (MR) brakes, namely two rotary MR brakes and one linear MR brake. The first rotary MR brake is integrated in the waist joint to reflect the horizontal tangent force, the other rotary MR brake is integrated in the shoulder joint to reflect the elevation tangent force, while the linear MR brake is integrated in the sliding joint of the arm to reflect the radial force (approach force). The proposed configuration can reflect a desired force to the operator at the end-effectors of the arm independently in 3 DOFs by controlling the current applied to the coils of the MR brakes. After the introduction, the configuration of the proposed force feedback system is presented. Afterward, the design and conducted simulation of the MR brakes for the systems are provided. The prototype of the force feedback system, which was manufactured for the experiment, is then presented as well as some of the obtained experimental results. Finally, the proposed control system is presented and its implementation to provide a desired feedback force to the operator is provided.

**Keywords:** MR fluid; spherical manipulator; force feedback system; rotary MR brake; linear MR brake

## 1. Introduction

Recently, much research has focused on the development and application of force feedback techniques with a wide range of applications, including medical and aerospace engineering, military operations, computer games, and virtual reality devices [1–4]. Several approaches have been proposed to build haptic systems using different types of actuators such as DC motors [5–7], electromagnetic actuators [8,9], pneumatic actuators [10,11], shape memory alloy [12], dielectric elastomeric actuators [13,14], eccentric rotating mass motors [15–17], and others.

Magnetorheological (MR) materials are a class of smart composites prepared by dispersing nanometer- or micrometer-sized ferromagnetic fillers into different carrier matrices. As the rheological properties can be controlled by an external magnetic field rapidly, reversibly, and continuously, magnetosensitive smart materials have great application potential in construction, automotive industry, artificial intelligence, etc. Depending on the carrier matrix material, different types of MR materials

can be composed such as MR fluid [18,19], MR elastomer [20,21], MR gel [22,23], MR foam [24,25] and MR grease [26,27]. In order to create a better desired feedback force, actuators featuring magnetorheological fluid (MRF) have been recently implemented [28–31]. These research studies showed that MRF-based actuators are very potential candidates for haptic applications. Recently, a 3-DOF haptic system featuring an anthropomorphic (elbow) manipulator and three MR brakes was developed by Nguyen et al. [32]. In the haptic tele-manipulator system developed by Nguyen et al., a commercial 3-DOF elbow manipulator was used as a slave manipulator while the haptic master manipulator had similar kinematic chain with the master, integrated three rotary MR brakes at the three joints to generate a desired force to the operator. The required torque at the joints of the haptic master was determined from the desired force based on static modeling of the manipulator. In order to achieve the required torque, a PID (Proportional-Integral-Derivative) controller was employed for each MR brake. Obviously, in the static modeling, the dynamic behavior of the manipulator was not accounted for. In addition, the friction was also neglected. Therefore, the calculated torques were not accurate required values, resulting in an inaccuracy of the feedback force. To improve the force feedback accuracy, Nguyen et al. [33] recently carried out some modifications, in which the required torques of the MR brakes were estimated based on the measured values of currents were applied to the armatures of the slave's servo motors. The experimental results showed that the accuracy of the feedback force was significantly improved compared to those presented in the case of [32]. An implicit disadvantage of a force feedback system using an articulated manipulator mechanism is that the required torques are determined from coupled equations, resulting in implicit values of the required torques. In order to solve the implicit torque calculation of the articulated manipulator-based force feedback, Nguyen el al. [34] proposed a force feedback system based on a spherical arm mechanism, in which two rotary MR brakes and a linear MR brake were employed to feedback a desired 3-DOF force to the operator. With the proposed configuration, a 3-DOF desired force acting on the master operator can be achieved by independently controlling the current applied to the coils of the MR brakes with this configuration.

It is to be noted that only the configuration of the force feedback system was proposed in [34], as well as some preliminary experimental works. This research is an extension of [34]. The rest of this paper is organized as follows. In Section 2, the configuration of the proposed 3-DOF force feedback system is presented. Afterward, in Section 3, the configurations of the three MR brakes for the systems are introduced and optimally designed. From the optimal results, a prototype of the force feedback system featuring the three optimized MR brakes is then manufactured and braking torques of the MR brakes are experimentally evaluated directly on the prototype force feedback system. In Section 4, a control system is proposed and implemented to generate a desired feedback force to the operator. In Section 5, experimental results on force feedback are obtained and shown with discussion. Finally, important remarks of this research work are summarized in Section 6.

## 2. The Proposed Spherical Force Feedback System

In this section, the configuration of the proposed 3-DOF force feedback system is shown in Figure 1. As can be seen, the system consists of a spherical arm mechanism with three joints—the waist revolute joint, the shoulder revolute joint, and the arm prismatic joint. On the shaft of the waist joint, a rotary MR brake is employed to reflect a desired horizontal tangent force. The housing of the brake is fixed to the body frame of the spherical arm, while its shaft is assembled to the joint shaft. On the other end of the brake shaft, an encoder is attached to measure the angular position of the body part of the master (azimuth angle). On the body part, the shoulder joint is installed. On the shaft of the shoulder joint, another rotary MR brake is installed to reflect a desired elevation tangent force. The housing of the brake is fixed to the body link while the brake shaft is connected to the shaft of the shoulder link. The other end of the brake shaft is connected with an encoder to measure the angular position of the arm link (elevation angle). The linear MR brake is installed on the shoulder link which is connected to the shaft of the shoulder joint. The outer housing of the linear MR brake is fixed inside the hole on

the shoulder link while the shaft can move back and forth as a sliding arm. At the end of the shaft, a 3-DOF force is employed to measure the force acting on the operator. In order to measure the position of the sliding arm (radial length), a linear encoder is used. The simplified kinematic chain of the force feedback system is shown in Figure 2.

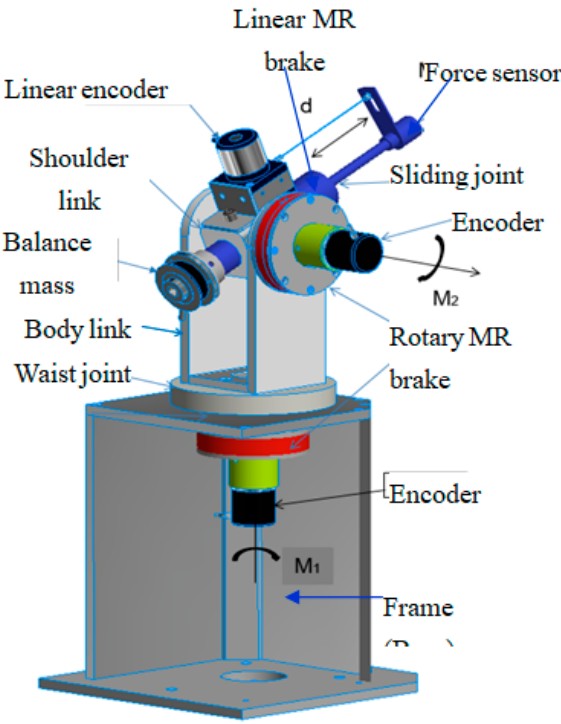

**Figure 1.** Configuration of the 3-DOF spherical force feedback system.

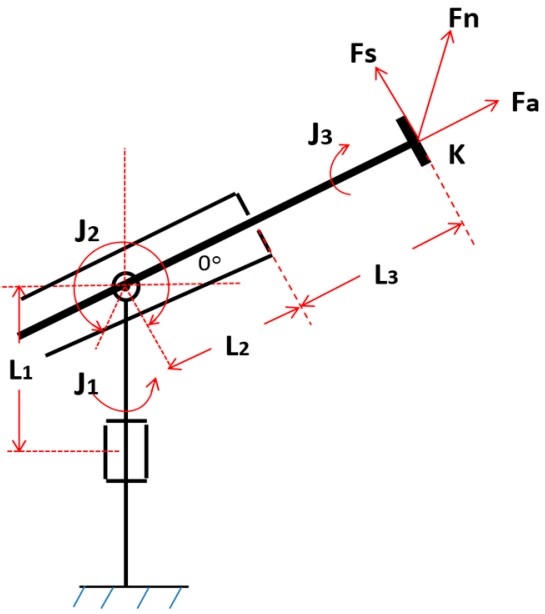

**Figure 2.** Kinematic chain of the 3-DOF force feedback system.

Figure 3 shows the fundamental dimensions of the force feedback system which are determined based on its required working space and manufacturing convenience. It is to be noted that the maximum reflected force in each direction (horizontal tangent force, elevation tangent force, approach force) is set

as 40 N considering the conformable operational effort of the operator. From these required forces, it can be determined that the maximum required torque of the rotary MR brakes is 8 Nm (=200 mm × 40 N), while the maximum required force of the linear MR brake is obviously 40 N.

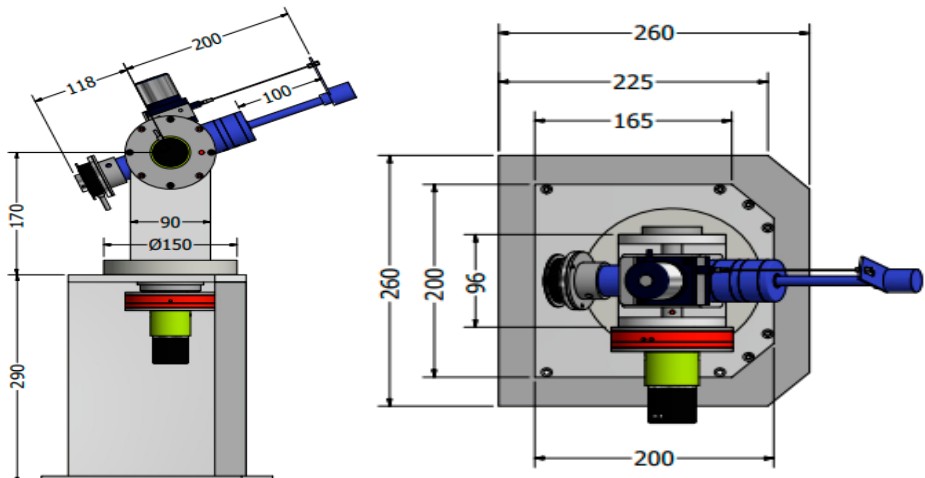

**Figure 3.** Significant dimensions of the force feedback system.

## 3. MR Brakes for the 3-DOF Force Feedback System

### 3.1. The Rotary MR Brakes

In this study, a novel MR brake with tooth-shaped rotor, shown in Figure 4, is proposed for the rotary MR brakes. As shown in Figure 4a, the brake disc (rotor) made of magnetic steel is fixed to the flange of the shaft made of nonmagnetic steel. The disc is installed inside a stationary envelope (housing). The housing is also made of magnetic housing. The gap between the disc and the housing is filled up with MR fluid. Two magnetic coils are placed on two sides of the brake housing to generate a magnetic field running across the MRF gap. It is to be noted that, to create a mutual magnetic field inside the brake structure, two counter currents are applied to the coils. The tooth-shaped counterpart faces of the disc and the housing allow a large contact surface between the MRF and the disc, thus the induced braking torque is expected to increase notably. Figure 4b shows simulated magnetic flux lines of the MR brakes obtained from ANSYS when two counter currents are applied to the coils.

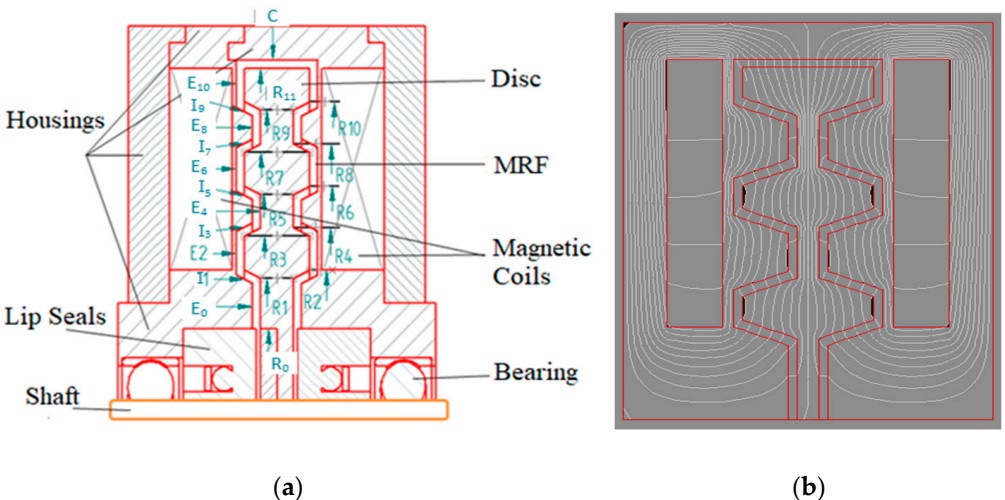

(**a**)　　　　　　　　　　　　　　　　　　　　　　　　(**b**)

**Figure 4.** Proposed rotary magnetorheological (MR) brakes. (**a**) Configuration of rotary MR brakes. (**b**) Simulated magnetic flux lines.

Based on the Bingham plastic model of the MR fluid, the braking torque of the brake is estimated by [35]:

$$T_b = 2(T_{E0} + T_{E2} + T_{E4} + T_{E6} + T_{E8} + T_{E10} + T_{I1} + T_{I3} + T_{I5} + T_{I7} + T_{I9}) + T_c + 2T_s \qquad (1)$$

where $T_{Ei}$ is the friction torque caused by MRF in the vertical gap $E_i$, $T_{Ii}$ is the friction torque caused by MRF in the inclined gap $I_i$, $T_c$ is the friction torque caused by MRF in the circular gap $C$, and $T_s$ is the lip seal friction force. The torque $T_{Ei}$, $T_{Ii}$ and $T_c$ are determined by the following:

$$T_{Ei} = \frac{\pi \mu_{Ei} R_{i+1}^4}{2d}\left[1 - \left(\frac{R_i}{R_{i+1}}\right)^4\right]\Omega + \frac{2\pi \tau_{yEi}}{3}(R_{i+1}^3 - R_i^3), \ i = 0, 2, 4, 6, 8, 10 \qquad (2)$$

$$T_{Ii} = 2\pi(R_i^2 l + R_i l^2 \sin\phi + \tfrac{1}{3}l^3 \sin^2\phi)\tau_{yIi} + \tfrac{1}{2}\pi\mu_{Ii}\tfrac{\pi}{d}(4R_i^3 + 6R_i^2 l \sin\phi + 4R_i l^2 \sin^2\phi + l^3 \sin^3\phi); \qquad (3)$$
$$i = 1, 3, 5, 7, 9$$

$$T_c = 2\pi R_{11}^2(b + 2h)\left(\tau_{yc} + \mu_c \frac{\Omega R_{11}}{d}\right) \qquad (4)$$

In the above, $R_i$ is the radius of the point $i$ in the disc profile as shown in Figure 4, $l$ is the length of the inclined gap, $\phi$ is the inclined angle, $h$ is the height of the tooth, $\mu_{Ei}$ and $\tau_{Ei}$ are the post-yield viscosity of the MRF in the gap $E_i$, $\mu_{Ii}$ and $\tau_{Ii}$ are the post-yield viscosity of the MRF in the gap $I_i$, and $\mu_c$ and $\tau_c$ are the post-yield viscosity of the MRF in the gap denoted by $C$. In this study, the induced yield stress and post-yield viscosity of MRF are functions of the applied magnetic density across the MRF duct, which are approximated by [36]:

$$Y = Y_\infty + (Y_0 - Y_\infty)(2e^{-B\alpha_{SY}} - e^{-2B\alpha_{SY}}) \qquad (5)$$

where $Y$ stands for the rheological parameters of MRF such as the yield stress and the post-yield viscosity. The value of $Y$ tends from the zero-applied field value $Y_0$ to the saturation value $Y_\infty$. $\alpha_{SY}$ is the saturation moment index of the $Y$ parameter. $B$ is the applied magnetic density. In this study, the commercial MRF made by Lord Corporation (Cary, North Carolina, United States), MRF132-DG, was used and its rheological properties according to Equation (5) are as follows [36]: $\mu_0 = 0.1$ Pa.s, $\mu_\infty = 3.8$ Pa.s, $\tau_{y0} = 15$ Pa, $\tau_{y\infty} = 40000$ Pa, $\alpha_{s\mu} = 4.5$ T$^{-1}$, and $\alpha_{st_y} = 2.9$ T$^{-1}$.

The lip seal friction torque can be approximately evaluated by the following:

$$T_s = 0.65(2R_s)^2 \Omega^{1/3} \qquad (6)$$

where $T_s$ is the friction torque of the seal measured in ounce-inches, $\Omega$ is the angular speed of the brake shaft measured in rounds per minute, and $R_s$ is the radius of the sealing shaft measured in inches.

It is well known that the braking torque and the mass of MR brakes are the two most important conflicting objectives in the design of MRF-based brakes, especially for force feedback systems. The brake mass should be as small as possible for compact size and low cost. However, the small size may reduce the brakes' braking torque. Therefore, in this research, the optimal design objective of the MR brake was to minimize the brake mass while its braking torque was constrained to be greater than a required value. Mathematically, the optimal design problem of the MR brakes can be stated as follows:

Minimize the MR brake mass

$$m_b = V_d \rho_d + V_h \rho_h + V_s \rho_s + V_{MR} \rho_{MR} + V_c \rho_c \qquad (7)$$

Subject to

$$\text{Braking torque constraint}: \ T_b \geq T_{br},$$
$$\text{Design variable limits}: \ x_i^L \leq x_i \leq x_i^U, \ i = 1, 2 \dots n.$$

where $V_d$, $V_h$, $V_s$, $V_{MR}$, and $V_c$ are respectively the volume of the disc, the housing, the shaft, the MRF, and the coil of the brake; $\rho_d$, $\rho_h$, $\rho_s$, $\rho_{MR}$, and $\rho_c$ are the density of materials for the discs, the housing, the shaft, the MRF, and the coil, respectively; $x_i^L$ and $x_i^U$ are the lower and upper bounds of the corresponding geometric design variable $x_i$ of MR brakes; $n$ is the number of design variables; and $T_{br}$ is the required braking torque. It is to be noted that in the optimization process, in order to evaluate the braking torque of the MR brakes, finite element analysis (FEA) is used to calculate magnetic flux density across the MRF gap. In this study, the finite element models using the axis-symmetric couple element (PLANE 13) of the commercial ANSYS software were used and are shown in Figure 5.

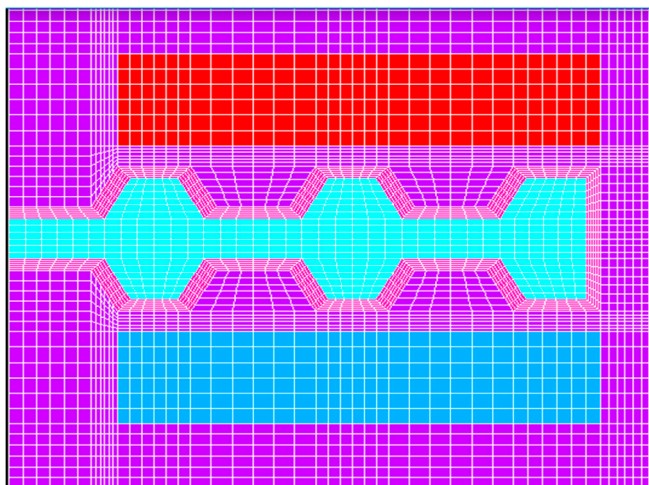

**Figure 5.** Finite element model for magnetic analysis of the rotary MR brakes.

The optimal solution of the MR brakes was obtained based on the first-order optimization method using the steepest gradient decent algorithm integrated in the ANSYS optimization toolbox [37,38]. In the optimal design problem, important parametric dimensions of the MR brakes such as the height and width of the coil ($h_c$, $w_c$), the disc outer radius $R_0$, the inner tooth radius $R_1$, the geometric dimensions of the tooth (height, top thickness, bottom thickness), the disc thickness $t_d$, the outer housing thickness $t_o$, and the side housing thickness $t_h$ were chosen as design variables. Noteworthily, a small gap size of the MRF duct results in a high braking torque. However, the small gap size causes a large value of the off-state braking torque that degrades performances of the MR brakes, such as high dissipated energy, overheating, and a particularly inaccurate feedback force due to the high value of the uncontrollable force. In addition, a small gap size of the MRF ducts results in cost and difficulty in manufacturing. important issues that should also be accounted for. Based on the above, the MRF gap size was not considered as a design variable but was empirically determined, being 0.6 mm in this research. The thin-wall thickness of the coil was also not considered as a design variable. It should be as small as possible to block magnetic flux going through but to force the magnetic flux going across the MRF gap. Considering manufacturing aspects, the thin-wall thickness of the coils was set as 1 mm. In this research work, the medium induced yield stress MRF made by Lord Corporation (MRF132-DG) was used and the copper coil wire was sized as 24-gauge (diameter = 0.511 mm, maximum working current is around 3 A). However, during the optimization process, a current of 2.5 A was applied to the coil to take safe working conditions into consideration. It is also to be noted that the filling ratio of the coil was set as 0.7, while magnetic loss was assumed to be 10% based on empirical experience.

The optimal solution of the MR brake, where the maximum induced braking torque was constrained to be greater than 10 Nm (the required is 8 Nm) and convergence rate was set as 0.1%, is shown in Figure 6a.

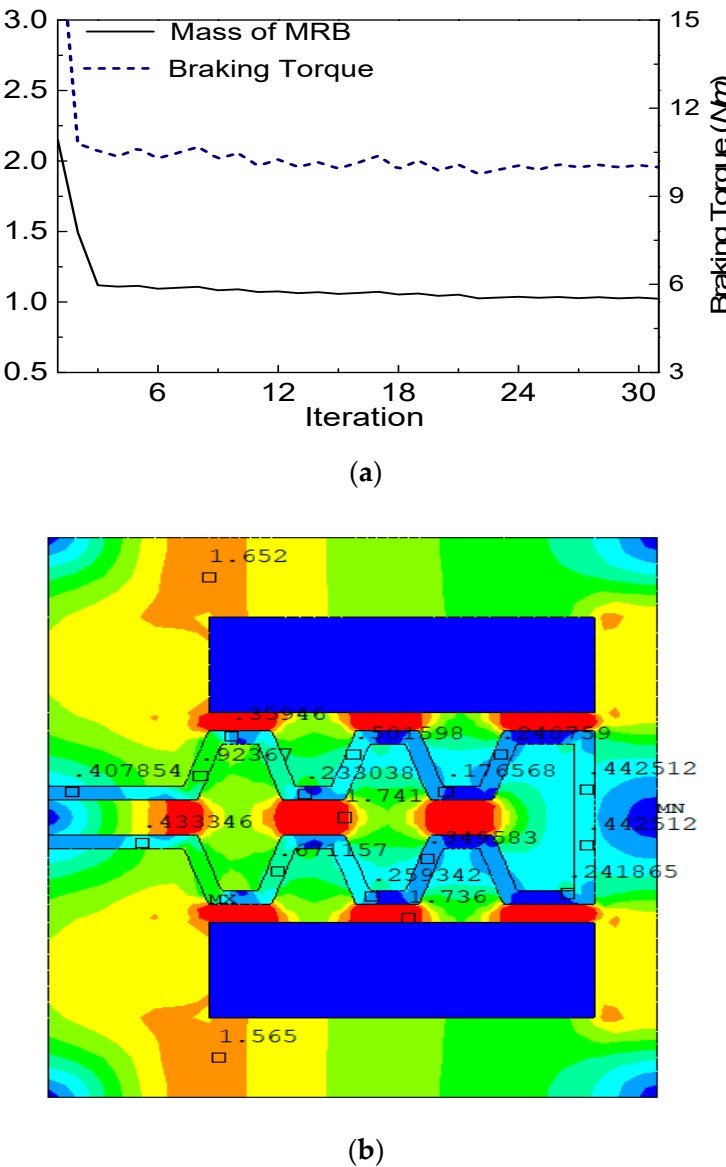

(a)

(b)

**Figure 6.** Optimization solution of the rotary MR brakes. (**a**) Mass and braking torque. (**b**) Magnetic flux density of the brake at the optimum.

From the figure, it can be seen that the convergence occurs after the 30[th] iteration, at which point the mass of the MR brake is 1.03 kg. The distribution of magnetic density is shown in Figure 6b. At the optimum, the values of design variables and performance of the MR brakes are shown in Table 1. It is to be noted in Table 1 that the thickness of the disc reached its lower limit at the optimum. From the optimal results, a detailed design of the MR brakes was carried out, and its assembly drawing is shown in Figure 7.

**Table 1.** Optimal results of the rotary MR brakes.

| Design Variable (mm) | Optimized Performance Characteristics |
|---|---|
| Size of coils: width $w_c$ = 5.52; height: $h_c$ = 15.8; no. of coil turns = 233 | Maximum braking torque: 10 Nm |
| Housing: Outer radius $R$ = 34.5; overall length $L$ = 35.8; thickness $t_h$ = 4.6; thin wall: 1.0 | Mass: 1.03 kg |
| Disc: Inner radius: $R_i$ = 10; outer radius: $R_d$ = 31.2; shaft radius = 6.0; thickness: 2.0 | Off-state torque: 0.1 Nm |
| Tooth profile: Total depth = 3.2; top thickness: 2.6, bottom thickness: 4.6 | Power consumption: 37 W |
| MRF gap: 0.8 | Coil resistant: 2.9 Ω |

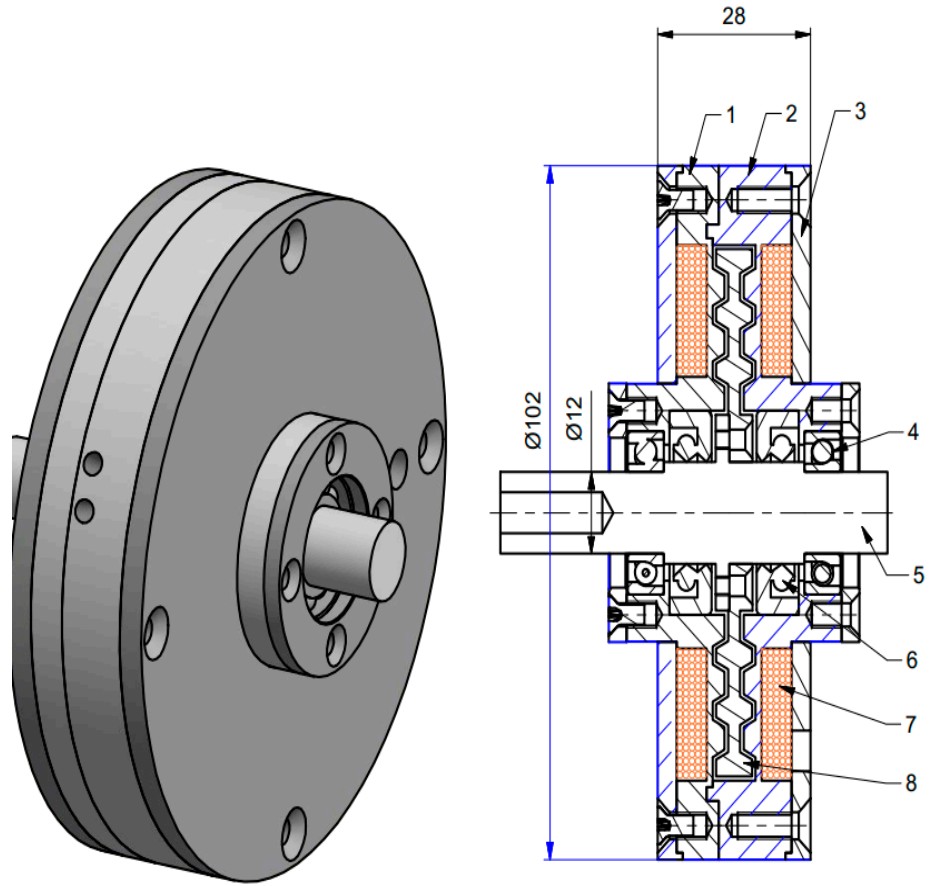

**Figure 7.** Assembly design of the rotary MR brakes.

### 3.2. The Linear MR Brake

In order to create a desired force in the radial direction, a two-coil linear MR brake without using a non-magnetic bobbin developed by Song et al. [39] shown in Figure 8 was used. The braking force of the linear MR brake can be determined by the following:

$$F_{sd} = 2\pi R_{sl}L(\tau_y + \mu\frac{v}{t_g}) + 2F_{or} = \frac{2\pi\mu R_{sl}Lv}{t_g} + 2(\pi R_{sl}L\tau_y + F_{or}) \tag{8}$$

where $R_{sl}$ is the radius of the brake shaft, $t_g$ is the MRF gap size, $v$ is the relative velocity between the shaft and the housing, $\mu$ and $\tau_y$ are average post-yield viscosity and yield stress of the activated MRF in the duct, $L$ is the length of the MRF duct, and $F_{or}$ is the friction force between the shaft and the O-ring which can be approximately calculated by the following:

$$F_{or} = f_c L_o + f_h A_r \tag{9}$$

where $L_o$ is the seal rubbing surface length (the shaft circumference), $f_c$ is the friction per unit length of the shaft circumference depending on the percentage of compression and the hardness of the O-ring material, $f_h$ is the friction force of the O-ring due to fluid pressure acting on a unit seal projected area, and $A_r$ is the seal projected area. For the linear brake used in this research, the pressure acting on the O-rings is very small and can be neglected and $f_h$ can be neglected. It is also to be noted that a 70-durometer rubber O-ring was used and the compression of O-ring was set as 7.5% in this study. From these data, the friction per unit length of the shaft circumference can be determined by $f_c =$ 87.5 N/mm.

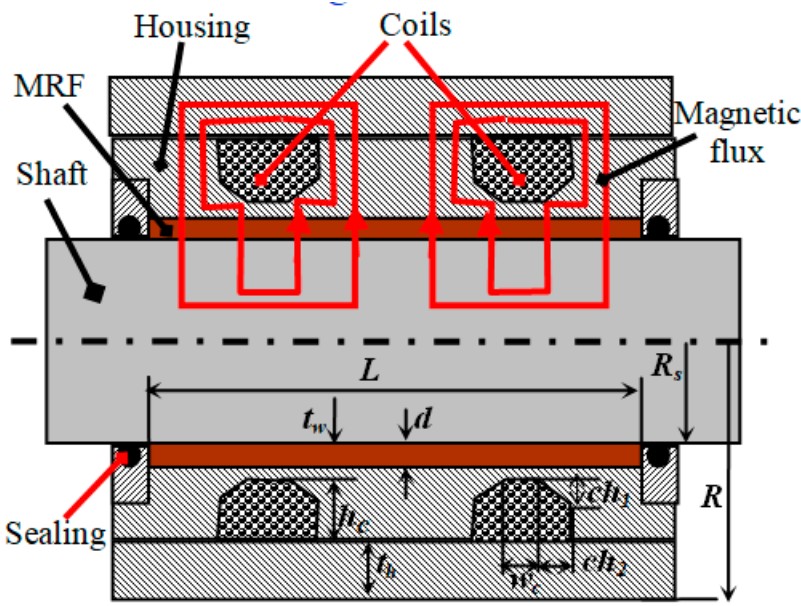

**Figure 8.** Configuration of the linear MR brake.

In the same way for the optimal design of the rotary MR brakes, the optimal design of the linear MR brake was conducted based on FEA. Figure 9 shows the finite element model used in this study for the linear MR brake. In this case, the coil height ($h_{cl}$), the coil width ($w_{cl}$), the chamfer geometry ($ch_1$ and $ch_2$), the MRF duct length $L$, the shaft radius $R_{sl}$ and the housing thickness $t_{hl}$ were chosen as design variables. The MRF gap size was empirically set as 0.8 mm, while the thin-wall thickness was set as 0.5 mm. The MRF132-DG and the 24-gauge copper wire were also used for this linear MR brake. It is to be noted that the off-state force of the linear MR brake should be as small as possible to have a more accurate reflection of the radial force. Therefore, the off-state force was assigned as an objective function in the optimization of the linear MR brake.

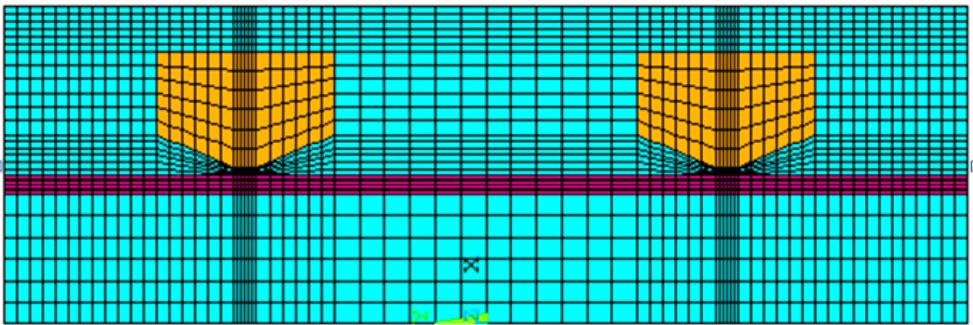

**Figure 9.** Finite element model for the magnetic analysis of the linear brake.

The optimal solution of the MR brake with a maximum required braking force of 40 N was achieved and is shown in Figure 10. From the figure, it can be seen that the convergence occurs after the 29th iteration, at which point the off-state force is 6.2 N. The maximum brake force is 40 N as required and the mass of the brake is 0.51 kg. The magnetic density distribution of the MR brake is shown in Figure 10b. At the optimum, the values of design variables and performance of the brake are summarized in Table 2. The detailed design of the MR brake at the optimum was then carried out, and its assembly drawing is shown in Figure 11.

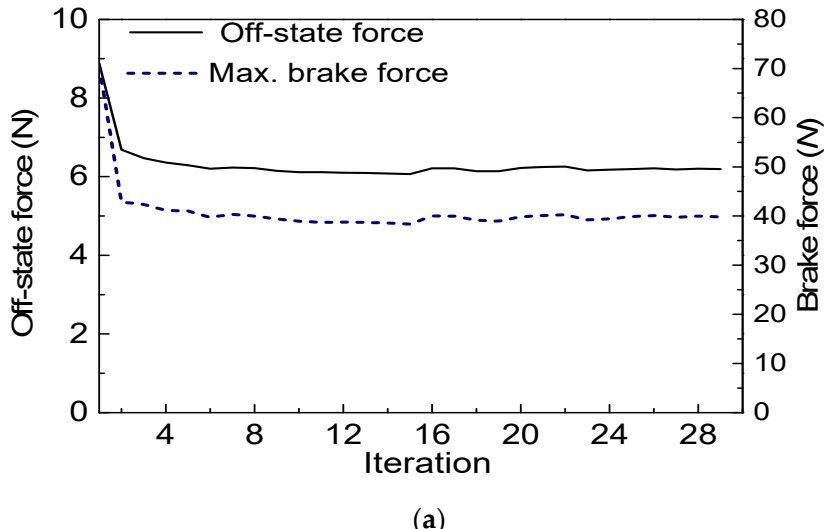

**(a)**

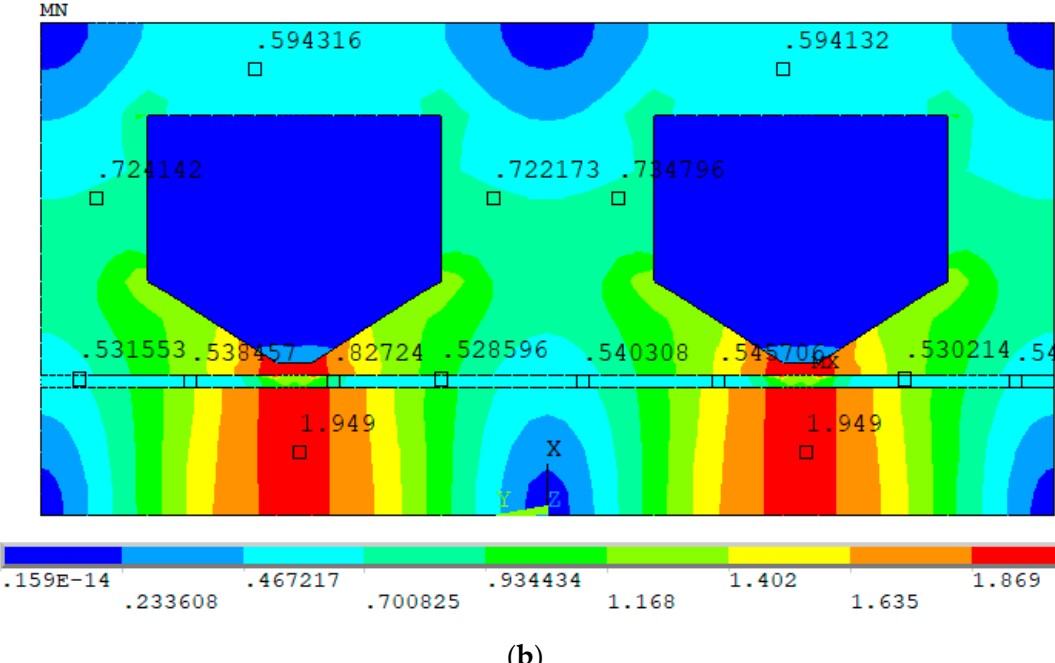

**(b)**

**Figure 10.** Optimization solution of the linear MR brake. (**a**) Mass and braking torque. (**b**) Magnetic flux density of the brake at the optimum.

**Table 2.** Optimal results of the linear MR brake.

| Design Variable (mm) | Optimized Performance Characteristics |
|---|---|
| Size of coils: width $w_{cl}$ = 1.5; height: $h_{cl}$ = 11.3; the chamfer: $ch_1$ = 3.7, $ch_2$ = 5.0, no. of coil turns = 386<br>Housing: Outer radius $R$ = 21.8; overall length $L$ = 39.2; thin wall: 0.5<br>Shaft: Shaft radius $R_{sl}$ = 5.0.<br>MRF gap: 0.6 | Maximum braking force: 40 Nm<br>Mass: 0.46 kg<br>Off-state torque: 6.0 N<br>Power consumption: 11.5 W<br>Coil radius: 2.5 Ω |

The three optimized MR brakes above were manufactured and installed on the spherical arm force feedback system as shown in Figure 12. It is to be noted that the real number of coil turns in the case of the rotary MR brakes is 230 turns (the calculated is 233 turns), whereas that of the linear MR brake is 380 for each coil (the calculated is 386 turns). In order to experimentally evaluate braking torque of the rotary MR brakes and braking force of the linear MR brake, experimental works were performed directly on the prototype.

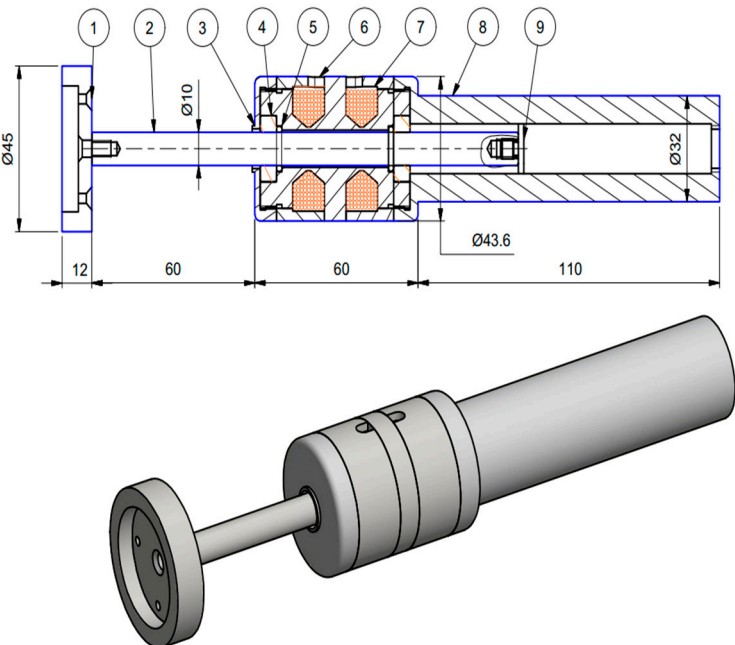

**Figure 11.** Assembly design of the linear MR brake.

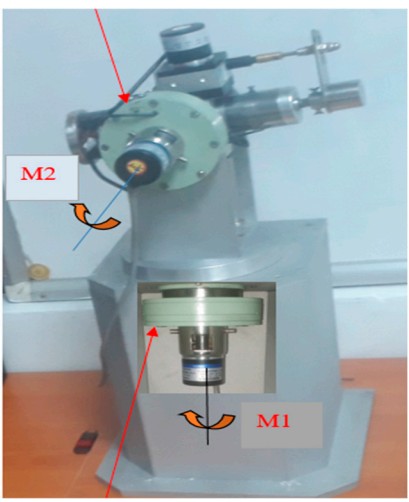

**Figure 12.** The 3-DOF spherical force feedback prototype.

Figure 13 shows the measured braking torque of the rotary MR brake at the waist joint. It is to be noted that, in the experiment, the horizontal tangent force at the end of the arm was measured by the 3-DOF force sensor and the arm was fixed in the horizontal direction with an arm length of 100 mm. Thus, the braking torque of the MR brake was calculated by the product of the measured force and the arm length (100 mm). During the experiment, the arm was rotated about the waist joint and the average value of the force at different values of the applied current was recorded. In the same manner of the MR brake at the waist joint, the braking torque of the MR brake at the shoulder joint was experimentally evaluated, and the results are shown in Figure 14. It is to be noted that, in this case, the arm was rotated about the shoulder joint axis during the experiment. Figure 15 shows the braking force of the linear MR brake at the function of the applied current. In this case, the arm was fixed in the horizontal direction and the shaft of the MR brake was moved back and forth.

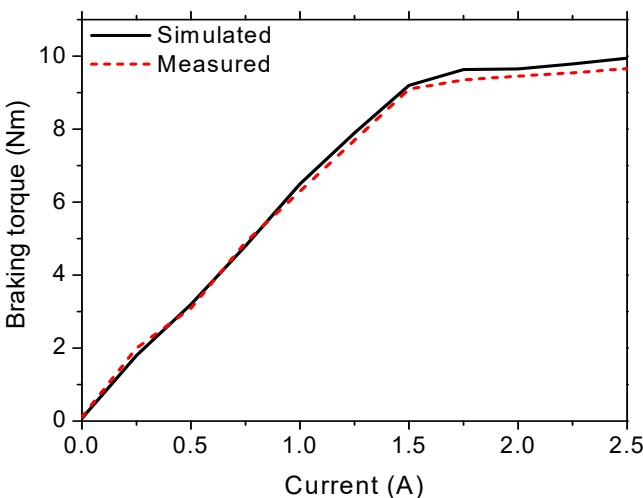

**Figure 13.** Braking torque of the waist MR brake vs. applied current.

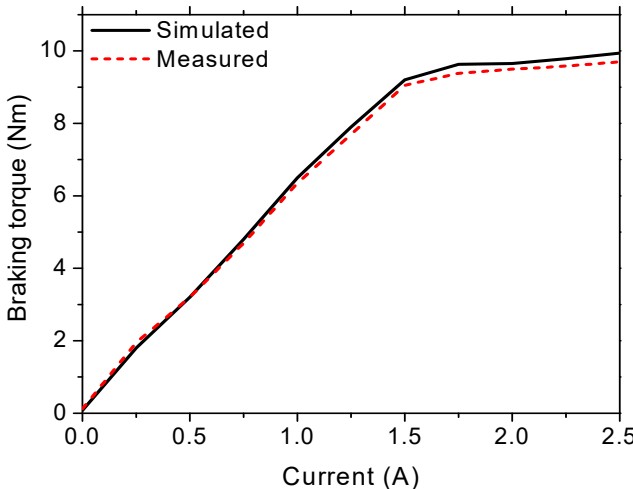

**Figure 14.** Braking torque of the shoulder MR brake vs. applied current.

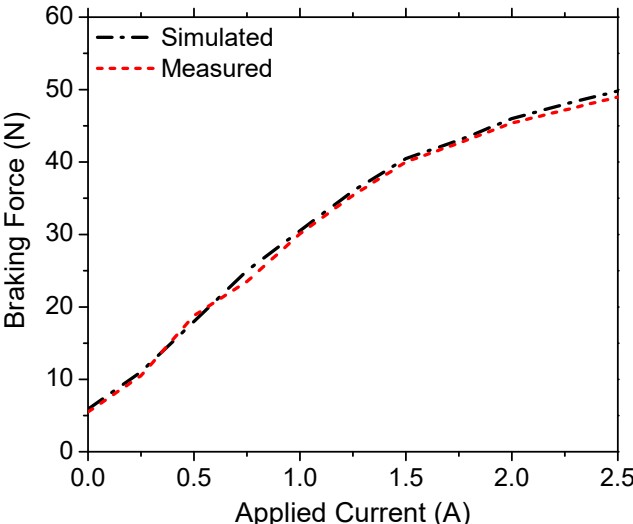

**Figure 15.** Braking force of the linear MR brake vs. applied current.

### 4. Control Design for the Force Feedback System

In this section, the open-loop controller is presented, which was designed to reflect a desired force to the operator. Figure 16 shows the block diagram of the control system for the horizontal and elevation tangent forces, whereas that for the radial force is shown in Figure 17. As shown in Figure 16, from the information of the encoders, the values of elevation angle ($\theta$) and arm radius $r$ were determined. The braking torque of the waist MR brake was then calculated by Equation (10), whereas that of the shoulder brake was calculated by Equation (11):

$$T_w = F_h r \cos \theta \tag{10}$$

$$T_{sh} = F_e r \tag{11}$$

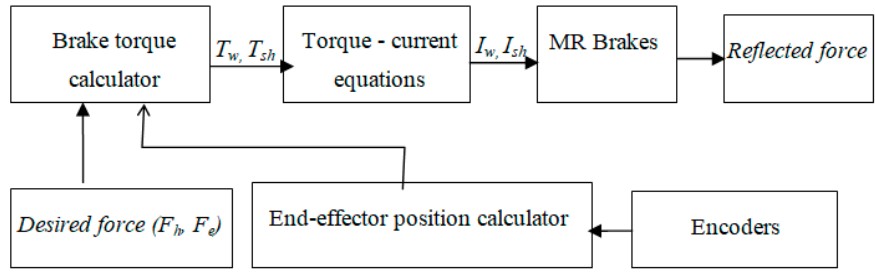

**Figure 16.** Block diagram of the control system for tangent force.

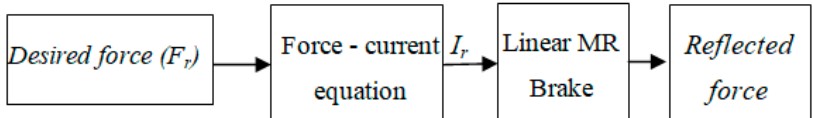

**Figure 17.** Block diagram of the control system for radius force.

In the above equations, $F_h$ is the desired horizontal tangent force and $F_e$ is the desired elevation tangent force. In order to achieve a braking torque equal to the calculated one, the applied current to the coils was determined from the experimental results shown in Figures 13 and 14. It can be seen that the braking torque of the MR brake was almost saturated when the applied current was greater than 1.5 A. Therefore, only applied current smaller than 1.5 A was used, and the applied current as a function of the generated torque is shown in Figures 18 and 19. Using parabolic curve fitting, the current applied to the coils of the waist and the shoulder MR brakes was calculated by Equations (12) and (13), respectively.

$$I_w = -0.0245 + 0.1516T_w + 0.00177T_w^2 \tag{12}$$

$$I_{sh} = -0.027 + 0.1543T_{sh} + 0.00155T_{sh}^2 \tag{13}$$

For the radial force, as shown in Figure 17, from the desired radial force, the applied current to the coils of the linear MR brake was determined from the experimental results shown in Figure 15. Similar to the rotary MR brake, only applied current smaller than 1.5 A was used for the linear MR brake, as shown in Figure 20. Using parabolic curve fitting, the current applied to the coils of the linear MR brake was determined by Equation (14):

$$I_r = -0.1707 + 0.03424F_r + 0.000169F_r^2 \tag{14}$$

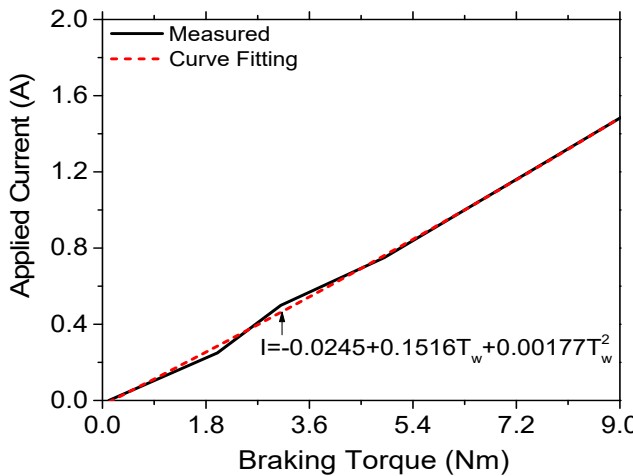

**Figure 18.** Applied current of the waist MR brake as a function of torque.

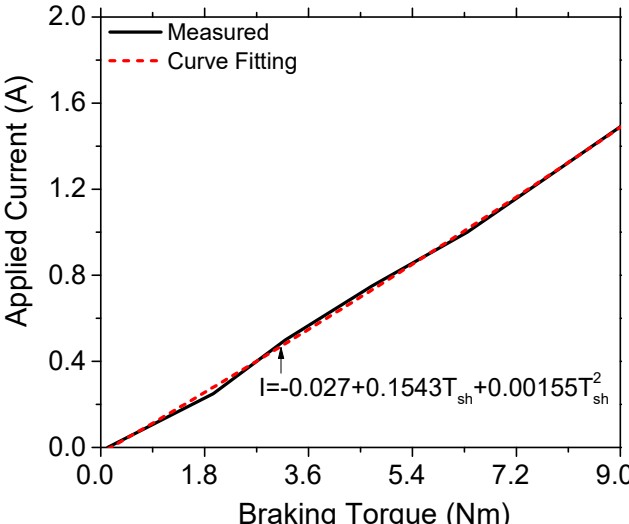

**Figure 19.** Applied current of the shoulder MR brake as a function of torque.

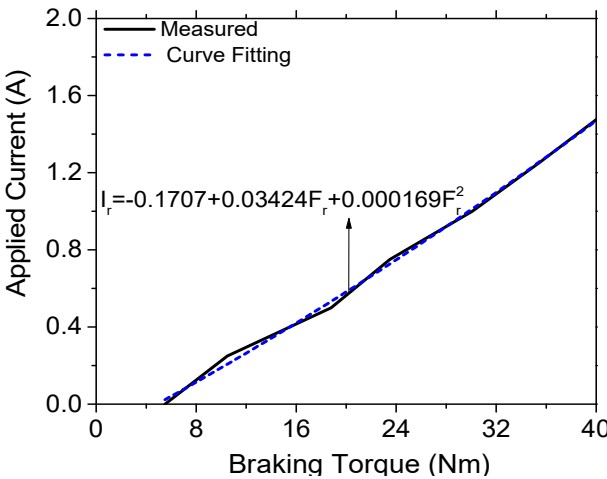

**Figure 20.** Applied current of the linear brake as a function of force.

## 5. Results and Discussion

Figure 21 shows the experimental setup for the proposed 3-DOF force feedback system. The interaction between the manipulator system and the computer and data acquisition was handled with a PCI card NI-6289 produced by National Instruments (Austin, Texas, United States and DSP system toolbox of MATLAB from MathWorrks, Inc. (Natick, Massachusetts, United States). A 3-DOF force sensor, OptoForce 3D OMD-20-FG-100N, produced by OptoForce Ltd. (Budapest, Hungary) was installed at the end of the arm to measure the reflected force. Angular positions of the master arm were measured by the rotary encoders, while the radial position was measured by the linear encoder. The information from the encoders was sent to the computer through the PCI card. From the encoders' information, the computer computed values of the applied currents for the MR brakes as mentioned in Section 4. It is to be noted that in this experimental setup, a voltage control signal ranging from 0 to 5 V from the computer was sent to an amplifier, that is, the Wonder Box Device Controller made by Lord Corporation. The output current, ranging from 0 to 2 A, was then applied to the coils of the MR brakes. In the experiment, the sample rate was set as 0.01 s.

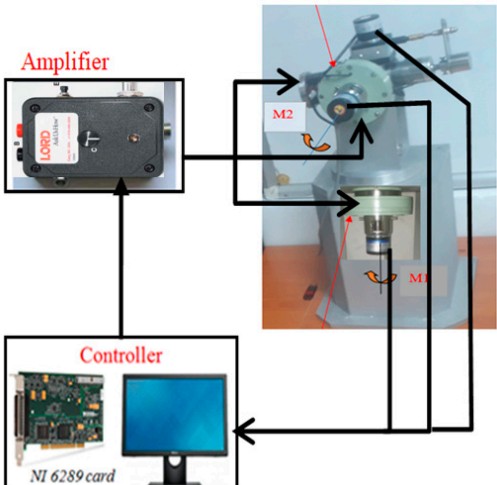

**Figure 21.** Experimental setup of the 3-DOF spherical force feedback system.

Figure 22 shows the step response of the force feedback system in which a constant desired feedback force of 40 N is set for each component of the feedback force (the horizontal tangent, the elevation tangent, and the radial force) at the time 0.5 s. In the experiment, the force feedback arm was located in an arbitrary position and the operator held the end-effector and moved it in an arbitrary trajectory in the working space during the experiment.

Figure 22a shows that the actual feedback horizontal force agrees well with the desired one, with a maximum error of 4% obtained at steady state and a response time of approximately 0.24 s. From Figure 22b, it is also be seen that the actual feedback elevation force agrees well with the desired one. In this case, the response time is approximately 0.26 s. From Figure 22c, it can be seen that the actual feedback radial force agrees quite well with the desired one, with a maximum error of approximately 6.5% and with more fluctuation than previous ones.

Figure 23 shows experimental results of the force feedback system. In the experiment, the force feedback arm was located in an arbitrary position; a sinusoidal desired feedback force was set for each component of the feedback force (the horizontal tangent, the elevation tangent, and the radial force). During the experiment, the operator held the end-effector and moved it in an arbitrary trajectory in the working space. It can be seen from Figure 23a that the actual feedback horizontal force agrees well with the desired one. However, the actual feedback force cannot be smaller than 1.5 N. It is obvious because of the uncontrollable torque (the off-state torque of the MR brake). From Figure 23b, it can also be seen that the actual feedback elevation force agrees well with the desired one. In this case,

the minimum achievable force is 1.8 N. Figure 23c shows that the actual feedback radial force agrees quite well with the desired one. However, in this case, the minimum achievable force is quite high, around 6 N, which is almost equal to the off-state force of the linear MR brake.

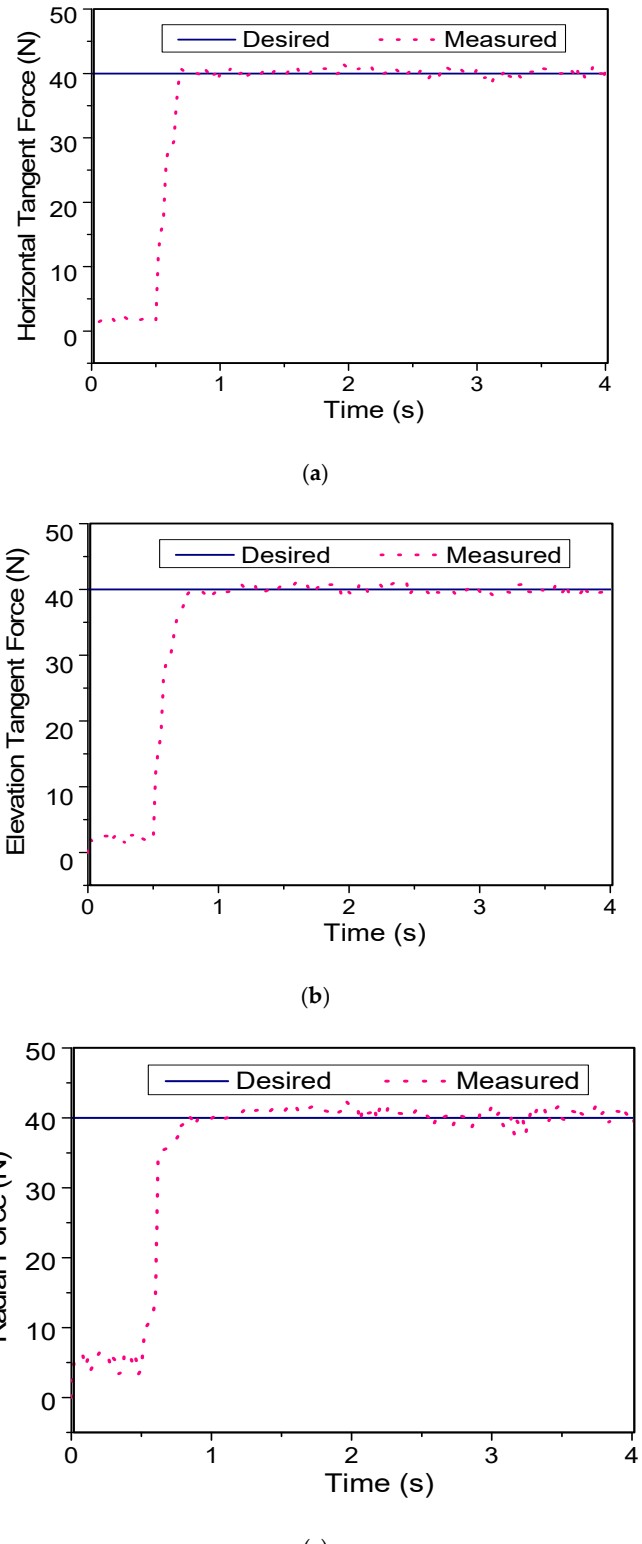

(a)

(b)

(c)

**Figure 22.** Experimental results on the step response of the force feedback system. (**a**) Horizontal force reflection. (**b**) Elevation force reflection. (**c**) Radial force reflection.

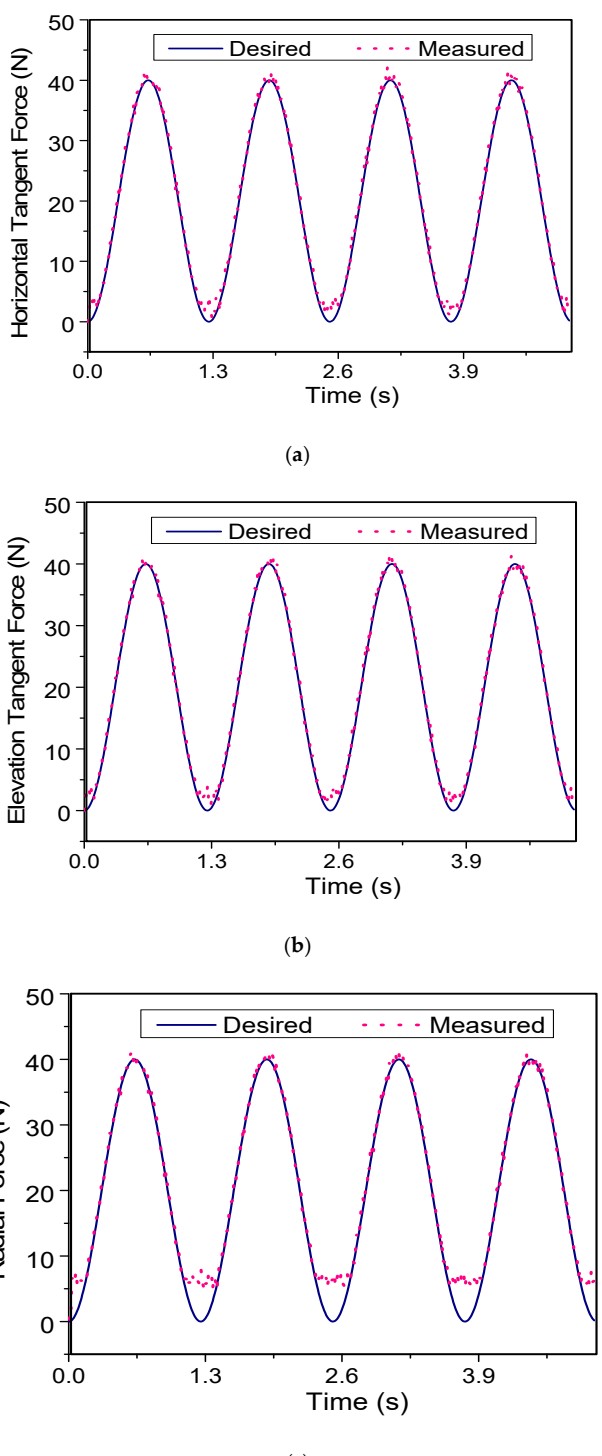

**Figure 23.** Experimental results on the sinusoidal response of the force feedback system. (**a**) Horizontal force reflection. (**b**) Elevation force reflection. (**c**) Radial force reflection.

The results showed that the proposed MRF-based 3-DOF haptic spherical manipulator can provide a good desired 3-DOF feedback force to the operator. In addition, by using the spherical arm mechanism, the force in each direction can be controlled independently and a closed-loop SISO (Single Input-Single Output) controller can be implemented to control the feedback force more accurately. It is to be noted that the proposed haptic spherical manipulator can be easily integrated with any slave robot for a haptic tele-manipulator system, in which a combination of force feedback in parallel with position control

is implemented. It is also noteworthy that the proposed master can be used as a universal master for all slave geometries (including identical or different kinematic chains). This leads to a potential application in modern situations such as tele-surgery, tele-operation in hazardous environment, etc. If the master and the slave have identical kinematic chains, a one-to-one copy of the joint motion can be used. On the other hand, in the case where the kinematic chain of the slave is different than that of the master, the position of the slave is controlled based on the position of the end effector of the master.

## 6. Conclusions

In this research work, a new 3-DOF force feedback system featuring a spherical arm mechanism and three MR brakes (two rotary MR brakes and one linear MR brake) was proposed. By using this proposed configuration, a 3-DOF desired force acting on the operator at the end-effector of the arm was achieved by independently controlling the current applied to the coils of the MR brakes. After the introduction, the configuration of the proposed force feedback system was presented. Afterward, the design and simulation of the MR brakes for the systems were conducted. For the rotary MR brake, a configuration of MR brake with a tooth-shaped rotor was employed. For the linear MR brake, a MR brake without using a non-magnetic bobbin was employed. In order to reflect a desired force, an open-loop controller based on the experimental performance of the MR brakes was used in this study. A prototype of the force feedback system was then manufactured for the experiment. Experimental results in the case of constant required feedback force showed that the time response of the force feedback is around 0.25 s and at steady state the maximum error of the horizontal tangent and the elevation tangent force is around 4%, while that of the radial force is up to 6.5% with more fluctuation. Experimental results with sinusoidal required feedback force show that a desired feedback force can be well achieved by the force feedback proposed system. However, due to the off-state torque and force of the MR brake, the system cannot reflect small force to the operator, which is 1.5 N for the horizontal fore, 1.8 N for the elevation force, and 6 N for the radial force.

As the second phase of this study, 3 close-loop SISO controllers will be implemented to control the feedback force more accurately. In addition, a tele-manipulator system using the proposed haptic spherical manipulator integrated with a slave manipulator will be conducted.

**Author Contributions:** B.T.D. is a PhD student working on this research topic. He is in charge of configuration design, detailed design, formal analysis, validation, prototype manufacturing and write the first draft of the manuscript; N.D.N. is in charge program coding and of data curation in this research; T.T.T. is responsible for planning and collecting resources, visualization, investiation and supervision of prototype manufacturing in this research; Q.H.N. is the corresponding author of this research work. He is in charge of conceptualization, methodology and final editing of the manuscript; All authors have read and agreed to the published version of the manuscript.

**Funding:** This work was supported by research funds from the Vietnam National Foundation for Science and Technology Development (NAFOSTED) under grant no. 107.01-2018.335 and partially by research funds from the Ministry of Education and Training of Vietnam under grant no. KYTH-114.

**Conflicts of Interest:** The authors declare no conflicts of interest.

## Nomenclature

| | |
|---|---|
| $A_r$ | Seal projected area |
| $B$ | Applied magnetic density across the MRF duct |
| $ch_1, ch_2$ | Chamfer geometry of the linear MR brake coil |
| $d$ | MRF gap size of the rotary MR brake |
| $F_e$ | Desired feedback elevation tangent force |
| $F_h$ | Desired feedback horizontal tangent force |
| $F_r$ | Desired feedback normal (radial) force |
| $F_{or}$ | Friction force between the shaft and the O-ring of the linear MR brake |
| $F_{sd}$ | Braking force of the linear MR brake |
| $f_c$ | Friction per unit length of the shaft circumference |
| $f_h$ | Friction force of the O-ring due to fluid pressure acting on a unit seal projected area |

| | |
|---|---|
| $h$ | Height of the tooth of the rotary MR brake |
| $h_c$ | Height of the MR brake coil |
| $h_{cl}$ | Height of the linear MR brake coil |
| $I_r$ | Current applied to the coils of the linear MR brake |
| $I_{sh}$ | Current applied to the coils of the shoulder MR brake |
| $I_w$ | Current applied to the coils of the waist MR brake |
| $L$ | Length of the MRF duct of the linear MR brake |
| $L_o$ | Seal rubbing surface length (the shaft circumference) |
| $l$ | Length of the inclined gap of the rotary MR brake |
| $R_i$ | Radius of the point $i$ in the disc profile of the rotary MR brake |
| $R_s$ | Radius of the MR brake shaft |
| $R_{sl}$ | Radius of the linear MR brake shaft |
| $R_0$ | Outer radius of the MR brake disc |
| $R_1$ | Inner tooth radius of the disc |
| $R_{sl}$ | Radius of the linear MR brake shaft |
| $r$ | Arm radius of the master manipulator |
| $T_b$ | Braking torque of the brake |
| $T_{br}$ | Required braking torque |
| $T_c$ | Friction torque caused by MRF in the circular gap $C$ of the rotary MR brake |
| $T_{Ei}$ | Friction torque caused by MRF in the vertical gap $E_i$ of the rotary MR brake |
| $T_{Ii}$ | Friction torque caused by MRF in the inclined gap $I_i$ of the rotary MR brake |
| $T_s$ | Friction force of the rotary MR brake seals |
| $T_{sh}$ | Calculated required torque of the shoulder MR brake |
| $T_w$ | Calculated required torque of the waist MR brake |
| $t_d$ | Thickness of the MR brake disc |
| $t_0$ | Thickness of the outer MR brake housing |
| $t_h$ | Thickness of the MR brake side housing |
| $t_{hl}$ | Thickness of the linear MR brake housing |
| $t_g$ | MRF gap size of the linear MR brake |
| $V_d$ | Volume of the disc of the MR brake |
| $V_h$ | Volume of the housing of the MR brake |
| $V_{MR}$ | Volume of the MRF of the MR brake |
| $V_s$ | Volume of the shaft of the MR brake |
| $V_c$ | Volume of the coil of the MR brake |
| $v$ | Relative velocity between the shaft and the housing of the linear MRR brake |
| $x_i^L$ | Lower bound of the corresponding geometric design variable $x_i$ |
| $x_i^U$ | Upper bound of the corresponding geometric design variable $x_i$ |
| $Y_0$ | Zero-applied field value of rheological parameter Y of MRF |
| $Y_\infty$ | Saturation value of rheological parameter Y of MRF |
| $w_c$ | Width of the MR brake coil |
| $w_{cl}$ | Width of the linear MR brake coil |
| $\alpha_{SY}$ | Saturation moment index of the rheological parameter Y of MRF |
| $\mu_{Ei}, \tau_{Ei}$ | Post-yield viscosity and induced yield stress of the MRF in the gap $E_i$ |
| $\mu_{Ii}, \tau_{Ii}$ | Post-yield viscosity and induced yield stress of the MRF in the gap $I_i$ |
| $\mu_c, \tau_c$ | Post-yield viscosity and induced yield stress of the MRF in the gap $C$ |
| $\mu$ | Average post-yield viscosity of the activated MRF of the linear MR brake |
| $\phi$ | Inclined angle of MRF gap of the rotary MR brake |
| $\Omega$ | Angular speed of the brake shaft |
| $\rho_d$ | Density of material of the MR brake disc |
| $\rho_h$ | Density of material of the MR brake housing |
| $\rho_{MR}$ | Density of material of the MR fluid |
| $\rho_s$ | Density of material of the MR brake shaft |
| $\rho_c$ | Density of material of the MR brake coils |
| $\tau_y$ | Average induced yield stress of the activated MRF of the linear MR brake |
| $\theta$ | Elevation angle |

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
