# Peer review of "Design and Experimental Validation of a 3-DOF Force Feedback System Featuring Spherical Manipulator and Magnetorheological Actuators†"

_actuators, doi:10.3390/act9010019_

Round 1

Reviewer 1 Report

The paper entitled Design and Experimental Validation of a 3-DOF Force Feedback System Featuring Spherical Manipulator and Magneto-rheological Actuators, authors Bao Tri Diep , Ngoc Diep Nguyen , Trung Thanh Tran , Quoc Hung Nguyen it is well structured

The authors investigate the development of a new 3-DOF force feedback system featuring a spherical arm mechanism and three magneto-rheological brakes (MR brakes).

The work brings added value to the scientific community.

A. The authors suggest that in the introductory part of the work must refer to Magnetorheological materials and its relevance regarding to properties, kind of materials used and behavior in MR brakes for 3-DOF force feedback systems. For this, the authors propose to consult and papers:

  1. Experimental Investigation of the Magnetorheological Behavior of PDMS Elastomer Reinforced with Iron Micro/Nanoparticles, Polymers 2017, 9, 696
  2. Fabrication and Characterization of Isotropic and Anisotropic Magnetorheological Elastomers, Based on Silicone Rubber and Carbonyl Iron Microparticles, Polymers 2018, 10, 1343
  3. Design and analysis of magnetorheological fluid brake for an all terrain vehicle,  IOP Conf. Series: Materials Science and Engineering 310 (2018) 012127  
  4. Experimental Performance Evaluation of a MR Brake-Based Haptic System for Teleoperation,  Experimental Performance Evaluation of a MR Brake-Based Haptic System for Teleoperation. Front. Mater. 6:25. doi: 10.3389/fmats.2019.00025
  5. Experimental Studies on Magnetorheological Brake for Automotive Application, International Journal of Automotive and Mechanical Engineering ISSN: 2229-8649 (Print); ISSN: 2180-1606 (Online) Volume 15, Issue 1 pp. 4893-4908 March 2018 
  6. Properties and application of magnetorheological fluids, Journal of Achievements in Materials and Manufacturing Engineering Volume 18 Issue 1-2 September 2006   

B. In Section 2 please provide some information about from data sheet of MRF used rotatory system

C. Some grammatical mistakes are present in the manuscript, please review all the paper and consult the rules of English language

Author Response

Point 1: The authors suggest that in the introductory part of the work must refer to Magnetorheological materials and its relevance regarding to properties, kind of materials used and behavior in MR brakes for 3-DOF force feedback systems. For this, the authors propose to consult and papers:

  1. 20 Experimental Investigation of the Magnetorheological Behavior of PDMS Elastomer Reinforced with Iron Micro/Nanoparticles, Polymers 2017, 9, 696
  2. 21 Fabrication and Characterization of Isotropic and Anisotropic Magnetorheological Elastomers, Based on Silicone Rubber and Carbonyl Iron Microparticles, Polymers 2018, 10, 1343
  3. Design and analysis of magnetorheological fluid brake for an all terrain vehicle,  IOP Conf. Series: Materials Science and Engineering 310 (2018) 012127  
  4. 28 Experimental Performance Evaluation of a MR Brake-Based Haptic System for Teleoperation,  Experimental Performance Evaluation of a MR Brake-Based Haptic System for Teleoperation. Front. Mater. 6:25. doi: 10.3389/fmats.2019.00025
  5. 19 Experimental Studies on Magnetorheological Brake for Automotive Application, International Journal of Automotive and Mechanical Engineering ISSN: 2229-8649 (Print); ISSN: 2180-1606 (Online) Volume 15, Issue 1 pp. 4893-4908 March 2018 
  6. 18 Properties and application of magnetorheological fluids, Journal of Achievements in Materials and Manufacturing Engineering Volume 18 Issue 1-2 September 2006   

Response 1: In the revised version, in the introduction part a brief introduction of Magnetorheological materials is added and five of the above suggested papers are cited (Lines 36-> 44, Ref. 18-28)

Point 2: In Section 2 please provide some information about from data sheet of MRF used rotatory system

Response 2: In the revised version, the information about from data sheet of MRF is mentioned (Lines 138-147).

Point 3: Some grammatical mistakes are present in the manuscript, please review all the paper and consult the rules of English language

Response 3: The languages have been checked again with the authors’s best efforts

Thank you very much

Reviewer 2 Report

Following the review of the paper “Design and Experimental Validation of a 3-DOF Force Feedback System Featuring Spherical Manipulator and Magneto-rheological Actuators”, I can conclude that:

The paper “Design and Experimental Validation of a 3-DOF Force Feedback System Featuring Spherical Manipulator and Magneto-rheological Actuators” is very interesting for industry, because it can contribute to developing a new generation of manipulators which uses the magneto-rheological brakes (MR brakes) and also force feedback system.

It can be specified several notable results such as:

The new solution for magneto-rheological brakes (MR brakes), rotary MR brake and linear MR brake, in terms of the computational models and practical models. The computational model showed good agreement with the experimental results; A prototype of the force feedback system is then manufactured for experiment; Finally, a control system is proposed and implemented to provide a desired feedback force to the operator.

In my opinion, this paper is interesting and shows a high degree of novelty as well as impact for high-tech industry.

But, the sections on the absolutely necessary aspects that are missing in the proposed version should be added in the paper as follows:

Because the paper refers to a lot of physical quantities (and most of them are explained in the text presented), a nomenclature which contains the physical quantities involved in the paper along with the measurement units in the international system, corresponding of the entire of the paper it would be desirable. It is necessary that the paper contains a nomenclature; The characterization data of the magneto-rheological liquid is missing in the paper. The paper must contain characterization data of the magneto-rheological fluid that is used in the proposed rotary MR brake and linear MR brake; To justify the use of magneto-rheological fluid, in the absence of a bibliography, the paper must contain the equations that describe the interaction of the magnetic field generated by the passage of electric current through the coils as a components of proposed rotary MR brake and linear MR brake, with the magneto-rheological fluid. Also, all the characteristics of these coils are not found. What is the diameter of the coils wire that was used? It is necessary to add in the paper a brief description of all logistics used for obtaining the Figure 21, “Experimental setup of the 3D spherical force-feedback system”, with indicate the all equipment being used in order to obtain the values for the experimental method and also the working way. What type of amplifier was utilized? Restore the Figure 21 in this respect would be desirable; The authors should improve in the paper the discussion regarding future practical deployment of their findings. This can be done into the conclusions part of the paper.

Author Response

Point 1: The characterization data of the magneto-rheological liquid is missing in the paper. The paper must contain characterization data of the magneto-rheological fluid that is used in the proposed rotary MR brake and linear MR brake; To justify the use of magneto-rheological fluid, in the absence of a bibliography, the paper must contain the equations that describe the interaction of the magnetic field generated by the passage of electric current through the coils as a components of proposed rotary MR brake and linear MR brake, with the magneto-rheological fluid.

Response 1: In the revised version, characterization data of the magneto-rheological fluid are added (Lines 138 -147)

Point 2: Also, all the characteristics of these coils are not found. What is the diameter of the coils wire that was used?

Response 2: In the revised version, characteristics of the coils are added (Lines 190, 191, 260->262)

Point 3: It is necessary to add in the paper a brief description of all logistics used for obtaining the Figure 21, “Experimental setup of the 3D spherical force-feedback system”, with indicate the all equipment being used in order to obtain the values for the experimental method and also the working way. What type of amplifier was utilized? Restore the Figure 21 in this respect would be desirable;

Response 3: In the revised version, more explanations are added for Figure 21(Lines 320 -> 330)

Point 4: The authors should improve in the paper the discussion regarding future practical deployment of their findings. This can be done into the conclusions part of the paper.

Response 4: In the revised version, the discussion regarding future practical deployment is added (Lines 356-361, 395->397)

Thank you very much

Round 2

Reviewer 2 Report

The paper “Design and Experimental Validation of a 3-DOF Force Feedback System Featuring Spherical Manipulator and Magneto-rheological Actuators”, in the revised form, is very interesting for industry, because it can contribute to developing a new generation of manipulators which uses the magneto-rheological brakes (MR brakes) and also force feedback system.

It can be specified several notable results such as:

  • The new solution for magneto-rheological brakes (MR brakes), rotary MR brake and linear MR brake, in terms of the computational models and practical models. The computational model showed good agreement with the experimental results;
  • A prototype of the force feedback system is then manufactured for experiment;
  • Finally, a control system is proposed and implemented to provide a desired feedback force to the operator.

In my opinion, this paper is interesting and shows a high degree of novelty as well as impact for high-tech industry.

Following the review of the paper The paper “Design and Experimental Validation of a 3-DOF Force Feedback System Featuring Spherical Manipulator and Magneto-rheological Actuators”, in the revised form, I can conclude that:

I think that the authors have solved correctly and thoroughly all the reviewer’s recommendations, except for the first recommendation: “Because the paper refers to a lot of physical quantities (and most of them are explained in the text presented), a nomenclature which contains the physical quantities involved in the paper along with the measurement units in the international system, corresponding of the entire of the paper it would be desirable. It is necessary that the paper contain a nomenclature”.

Also, have been introduced within the paper the result paragraphs. The paper is very interesting and shows a high degree of novelty and future impact.

In my opinion, the manuscript has been significantly improved and now warrants publication in Journal of Actuators in this form, without further revision.